# A novel approach to measure complex V ATP hydrolysis in frozen cell lysates and tissue homogenates

Lucia Fernandez-del-Rio[1,2] , Cristiane Benincá[1,2], Frankie Villalobos[1,2], Cynthia Shu[1,2] , Linsey Stiles[1,2,3], Marc Liesa[1,2,4,5] , Ajit S Divakaruni[2,3], Rebeca Acin-Perez[1,2] , Orian S Shirihai[1,2,3,4]

**Mitochondrial depolarization can initiate reversal activity of ATP synthase, depleting ATP by its hydrolysis. We have recently shown that increased ATP hydrolysis contributes to ATP depletion leading to a maladaptation in mitochondrial disorders, where maximal hydrolytic capacity per CV content is increasing. However, despite its importance, ATP hydrolysis is not a commonly studied parameter because of the limitations of the currently available methods. Methods that measure CV hydrolytic activity indirectly require the isolation of mitochondria and involve the introduction of detergents, preventing their utilization in clinical studies or any high-throughput analyses. Here, we describe a novel approach to assess maximal ATP hydrolytic capacity and maximal respiratory capacity in a single assay in cell lysates, PBMCs, and tissue homogenates that were previously frozen. The methodology described here has the potential to be used in clinical samples to determine adaptive and maladaptive adjustments of CV function in diseases, with the added benefit of being able to use frozen samples in a high-throughput manner and to explore ATP hydrolysis as a drug target for disease treatment.**

## Introduction

In energy-demanding tissues, ATP, the molecular unit in charge of intracellular energy transfer, is mainly produced by glycolysis and mitochondrial oxidative phosphorylation. Inside the mitochondria, the ATP synthase (complex V, CV) uses the electrochemical proton gradient generated by the electron transport chain and rotates clockwise to synthesize ATP using ADP and phosphate (Yoshida et al, 2001; Capaldi & Aggeler, 2002; Nath, 2002; Jonckheere et al, 2012). However, under certain conditions, when the electrochemical proton gradient declines, CV can reverse the direction of rotation to counterclockwise and hydrolyze ATP while actively transporting protons from the mitochondrial matrix into the intermembrane space (Capaldi & Aggeler, 2002; Dittrich et al, 2003; Gao et al, 2005; Li et al, 2015). The extrusion of protons by the reversal of ATP synthase rotation allows the proton gradient across the inner mitochondrial membrane to be maintained and thus prevents depolarization (Junge et al, 2009; Nicholls & Ferguson, 2013; Walker, 2013).

ATP hydrolysis is regulated both by the proton motive force (PMF) and by structural elements. PMF generated by the membrane potential and the proton gradient will determine the direction of synthesis or hydrolysis. Respiratory chain impairment can result in a drop in the PMF, which would result in ATP hydrolysis. In addition, several recent reports demonstrate that the membrane potential along the mitochondrial inner membrane is not homogenous (Wolf et al, 2019; Salewskij et al, 2020), suggesting that some CV particles within a mitochondrion may be exposed to conditions favoring hydrolysis, even in an apparently polarized mitochondrion. Individual cristae within the same mitochondrion may be depolarized, and monomers of CV that translocate to the lower potential inner boundary membrane may also see lower PMF.

Structural elements such as the mitochondrial matrix peptide ATPIF1 have been shown to prevent reversal activity of CV (Cabezon et al, 2001; Garcia-Bermudez & Cuezva, 2016; Garcia-Aguilar & Cuezva, 2018; Martin-Jimenez et al, 2018). ATPIF1 expression varies within tissues and pathologies. As such, the potential to engage in ATP hydrolysis is a derivative of a complex combination of biophysical parameters. As a result of CV ATP hydrolysis, cellular ATP depletion can occur and lead to severe defects when ATP is unavailable for key cellular processes (St-Pierre et al, 2000; Nieminen, 2003; Chinopoulos & Adam-Vizi, 2010). In accordance with this, we have recently demonstrated that an increase in mitochondrial ATP hydrolysis, rather than decreased synthesis, contributes to pathogenesis in models of primary and secondary mitochondrial deficiencies (paper under review in EMBO J, EMBOJ-2022-111699).

To fully understand the implications of the ATP hydrolysis in health and disease, we need to be able to measure its activity in a standardized, direct, and efficient way. However, currently, there is a gap between the existing and the necessary technology to address

[1]Department of Medicine, Endocrinology, David Geffen School of Medicine, University of California, Los Angeles, CA, USA   [2]Metabolism Theme, David Geffen School of Medicine, University of California, Los Angeles, CA, USA   [3]Department of Molecular and Medical Pharmacology, University of California, Los Angeles, CA, USA   [4]Molecular and Cellular Integrative Physiology, University of California, Los Angeles, CA, USA   [5]Institut de Biologia Molecular de Barcelona, IBMB-CSIC, Barcelona, Spain

Correspondence: oshirihai@mednet.ucla.edu; rbkacin@gmail.com

ATP hydrolytic capacity that includes the following: (1) the absence of a direct determination of the activity instead of an indirect assessment using linked reactions (Sumner, 1944; Pullman et al, 1960; Gomez-Puyou et al, 1983; Acin-Perez et al, 2008; Haraux & Lombes, 2019; Galber et al, 2021); (2) limitations in the kind of samples that can be tested, primarily isolated mitochondria with a few exceptions (Gomez-Puyou et al, 1983; Divakaruni et al, 2018b; Haraux & Lombes, 2019; Galber et al, 2021); (3) the lack of standardized sample preparation; (4) in some cases, the use of detergents, which may influence the final readout; or (5) a combination of the previously mentioned issues. Some of these limitations are particularly noticeable for cells in culture, especially primary cells derived from patients or other clinical samples, where the preparation of isolated mitochondria is challenging and the final amount of sample is limited. Furthermore, these limitations make it virtually impossible to use the current methodology to conduct high-throughput studies.

Here, we describe the development of a standardized, cost-effective, and widely applicable method to measure maximal ATP hydrolytic capacity per CV as a direct readout of hydrolytic activity. Our method is able to measure ATP hydrolytic capacity and maximal respiratory activity in the same assay, using previously frozen samples with minimal disruption (i.e., no addition of detergents). We called this method HyFS, which stands for hydrolysis in frozen samples. The method is optimized to be used with frozen lysates obtained from primary patient-derived fibroblasts, but it is also compatible with previously frozen PBMCs, tissue homogenates, and isolated mitochondria from different sources. Furthermore, the utilization of the Agilent Seahorse XF96 technology allows for relatively high-throughput studies. Our methodology has the potential to expand our knowledge around the extent of hydrolytic activity of CV under normal conditions and in pathology to better understand the contribution of this phenomenon in the field of mitochondrial biology and mitochondrial diseases.

## Results

### Measuring mitochondrial CV ATP hydrolytic capacity

ATP hydrolysis by CV is controlled both by structural elements of the complex and by the mitochondrial PMF (Nicholls & Budd, 2000; Nicholls, 2002; Duchen, 2004). Although data on mitochondrial membrane potential are available for a large set of disease models, the differences in maximal hydrolytic capacities per CV are rarely available and the methods by which these data were collected have not been standardized. Our goal was to establish an approach that will allow for standardized comparison of the CV ATP hydrolytic capacities from tissue samples and cell samples collected from different disease models. When working in reverse mode, CV catalyzes a reaction that cleaves ATP to ADP + P. As a byproduct of the reaction, approximately one proton is released per two molecules of ATP hydrolyzed (Hochachka & Mommsen, 1983; Robergs et al, 2004). The release of protons can be noted as a drop in pH, and monitored using the extracellular acidification rate (ECAR) in a Seahorse XF96 Extracellular Flux analyzer (McQuaker et al, 2013;

Divakaruni et al, 2018a; Acin-Perez et al, 2021). Changes in pH can also be the result of $CO_2$ release and lactate production due to glycolytic turnover (Divakaruni et al, 2014; Divakaruni et al, 2018a) that can be observed when using intact cells. We rationalized that the drop in pH associated with the release of $CO_2$ or lactate could be eliminated by breaking cellular and mitochondrial membranes. Breaking the cellular and mitochondrial membranes will block glycolysis and the $CO_2$ generated through the Krebs cycle, removing their potential contributions to acidification rate (AR). Therefore, in previously frozen isolated mitochondria, frozen tissues, or frozen lysates, pH variations would constitute a direct readout of ATP hydrolytic capacity by CV. For accuracy, the ECAR channel has been retermed as "AR" channel, removing the extracellular classification, as intact cells are not used for this method.

To monitor ATP hydrolytic capacity of CV, the electron transport chain must be inhibited to prevent any contribution of the forward rotation of CV. Then, when ATP is provided, CV will change to its reverse mode (Fig 1). The $H^+$ generated by ATP hydrolysis will acidify the experimental medium over time (t) (Divakaruni et al, 2014; Divakaruni et al, 2018a), concomitant to an increase in ΔpH/Δt, which represents an increase in the AR. In our assay, the ATP is provided in combination with FCCP, to dissipate any remaining membrane potential and maintain ATP hydrolysis as the favorable reaction. Finally, oligomycin is injected to completely inhibit CV in both the forward and reserve directions. Sensitivity to oligomycin indicates the proportion of acidification that is directly linked to CV. To obtain a side-by-side analysis of hydrolytic capacity and maximal respiratory capacity, we accompanied the hydrolysis assay with a measurement of maximal oxygen consumption as an additional parameter in the same run. The maximal activity of either CI or CII can be measured by providing the appropriate substrates to the previously frozen samples. In this way, the same sample will deliver information about both maximal respiratory activity and maximal ATP hydrolytic capacity. Because inhibition of respiration is required to measure acidification, maximal oxygen consumption is measured first, followed by the measurement of hydrolytic capacity (Fig 1).

The HyFS assay proceeds as follows: (1) samples are supplied with the suitable substrates, and oxygen consumption rate (OCR) is monitored (Figs 1 and 2A, top). (2) After 8 min, antimycin A (AA) is injected and respiration is blocked because of complex III inhibition, forcing CV into its hydrolytic mode in the presence of ATP. The remaining balance of (OCR before AA) − (OCR after AA) represents the substrate-driven respiration of the sample of interest (Figs 1 and 2A, top). Maximal respiration was fueled by succinate plus rotenone (SR), which fuel CII and inhibit CI, respectively. We will refer to this parameter as "SR respiration." In addition, NADH could be used to drive respiration when information on maximal respiration through complex I is of interest. (3) At minute 18, ATP + FCCP are added by injection, allowing CV reversal activity. Although we expect all proton gradients to be dissipated by the freeze and thaw, FCCP is still added as an additional assurance that membrane potential is collapsed. The H+ generated by ATP hydrolysis acidifies the medium, and the increase in acidification is observed in the ECAR channel (Figs 1 and 2A, bottom). (4) At minute 34, oligomycin is injected to completely block CV activity and the AR dramatically drops. The maximal ATP hydrolytic capacity by CV of a given sample

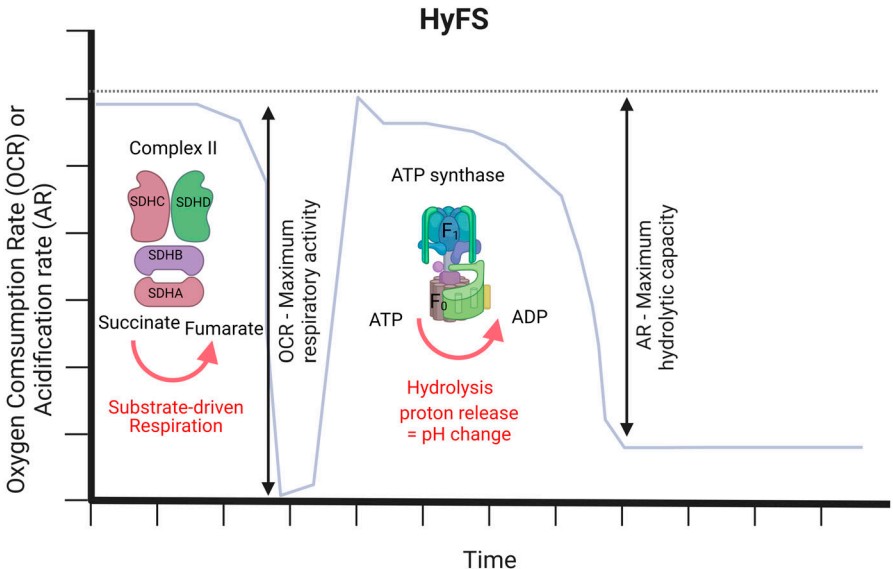

**HyFS**

**Figure 1. Rationale of the hydrolysis in frozen sample method.**
The method starts with a readout of the substrate-driven respiration of CII, which is completely inhibited before the addition of ATP. The addition of ATP will force CV to work in reverse to hydrolyze ATP. The proton released from the hydrolysis will acidify the medium, and an increase in acidification will be recorded as a direct readout of the activity. Oligomycin stops CV rotation setting the sensitivity of the assay. Source data are available for this figure.

is calculated as (AR signal after ATP) – (AR signal after oligomycin) (Figs 1 and 2A, bottom).

## Determination of CV hydrolytic capacity is more efficient in previously frozen mitochondria isolated from primary fibroblasts

With the above rationale, we started the assay development process with isolated mitochondria (Divakaruni et al, 2018a; Acin-Perez et al, 2021). We tested our method first in freshly isolated heart mitochondria. In Fig 2B, we show state 3 respiration, sustained by pyruvate/malate/ADP or succinate/rotenone/ADP (Fig 2B, top). After measuring respiration, we followed with the measurement of ATP hydrolytic capacity in the same sample (Fig 2B, bottom, and Fig S1A). The low levels of OCR in the samples pre-treated with oligomycin indicate that prepared mitochondria are intact, coupled, and able to synthesize ATP (Fig 2B, top). Because the phosphorylation of ADP alkalinizes the experimental medium, negative values of AR, indicating alkalinization, support that most of the ATP synthase was working in the synthesis rather than the hydrolysis mode, which was prevented by the addition of oligomycin (Fig 2B, bottom, and Fig S1A). Remarkably, ATP hydrolytic capacity was influenced by the respiratory substrate added during the test of respiration that preceded the hydrolysis test. In fresh heart mitochondria, we find that ATP hydrolytic capacity rate is higher if we started the assay with respiration on SR as compared to PM (Fig 2B, bottom, and Fig S1A). Because mitochondria generally consume oxygen faster with succinate and rotenone than with pyruvate and malate, we opted to use succinate to achieve a higher dynamic range in our future assays. Alternatively, complex I maximal respiration using NADH in previously frozen samples (Acin-Perez et al, 2020a) can be also measured when sample size is not limiting or in situations where measuring complex I might be of more interest than assessing complex II respiration through succinate/rotenone. To verify our method, we repeated it in fresh mitochondria isolated from primary human control fibroblasts (Fig 2C). Mitochondria are coupled and

active as shown in their readout of state 3 respiration (Fig 2C, top), but the increase in acidification is very low and the subsequent inhibition by oligomycin is minimal (Fig 2C, bottom, and Fig S1C). The poor sensitivity to oligomycin observed in these mitochondria may indicate the presence of other ATP hydrolases different from CV.

In an attempt to achieve a better ATP hydrolytic capacity signal and to eliminate the need for fresh samples, we decided to test our method in previously frozen mitochondria. Freezing and thawing partially disrupt mitochondrial membranes, allowing the enzymes a better access to the substrates. Using previously frozen mitochondria isolated from mouse heart, we quantified maximal CII respiration, or SR respiration, and ATP hydrolytic capacity in a concentration-dependent manner (Figs 2D and S1B) without losing specificity. In fact, the signal-to-noise ratio was improved (less noise) and a better dynamic range observed. To further assess the sensitivity to oligomycin of previously frozen heart mitochondria, we performed a titration using different doses of oligomycin, observing that the AR signal after ATP showed a dose-dependent sensitivity to oligomycin (Fig 2E and F). Moving to previously frozen mitochondria isolated from human primary fibroblasts, we could also obtain a good readout of CII maximal activity (Fig 2G, top) and, more importantly, a higher acidification signal followed by a stronger inhibition by oligomycin (Fig 2G, bottom, and Fig S1D) (Divakaruni et al, 2018a).

## Multiple freeze–thaw cycles increase the sensitivity of ATP hydrolytic capacity measurement in previously frozen cell lysates

To adapt the assay to clinical settings and to gain a higher throughput, we aimed to optimize the method to allow for the measurement of ATP hydrolytic capacity in previously frozen tissue homogenates and cell lysates. In Fig 3A, we show the measurement of CV ATP hydrolytic capacity in previously frozen murine heart homogenate. We confirmed that acidification in tissue homogenates represented mitochondrial CV hydrolytic activity, by its sensitivity to oligomycin and by comparing with cytosolic fractions,

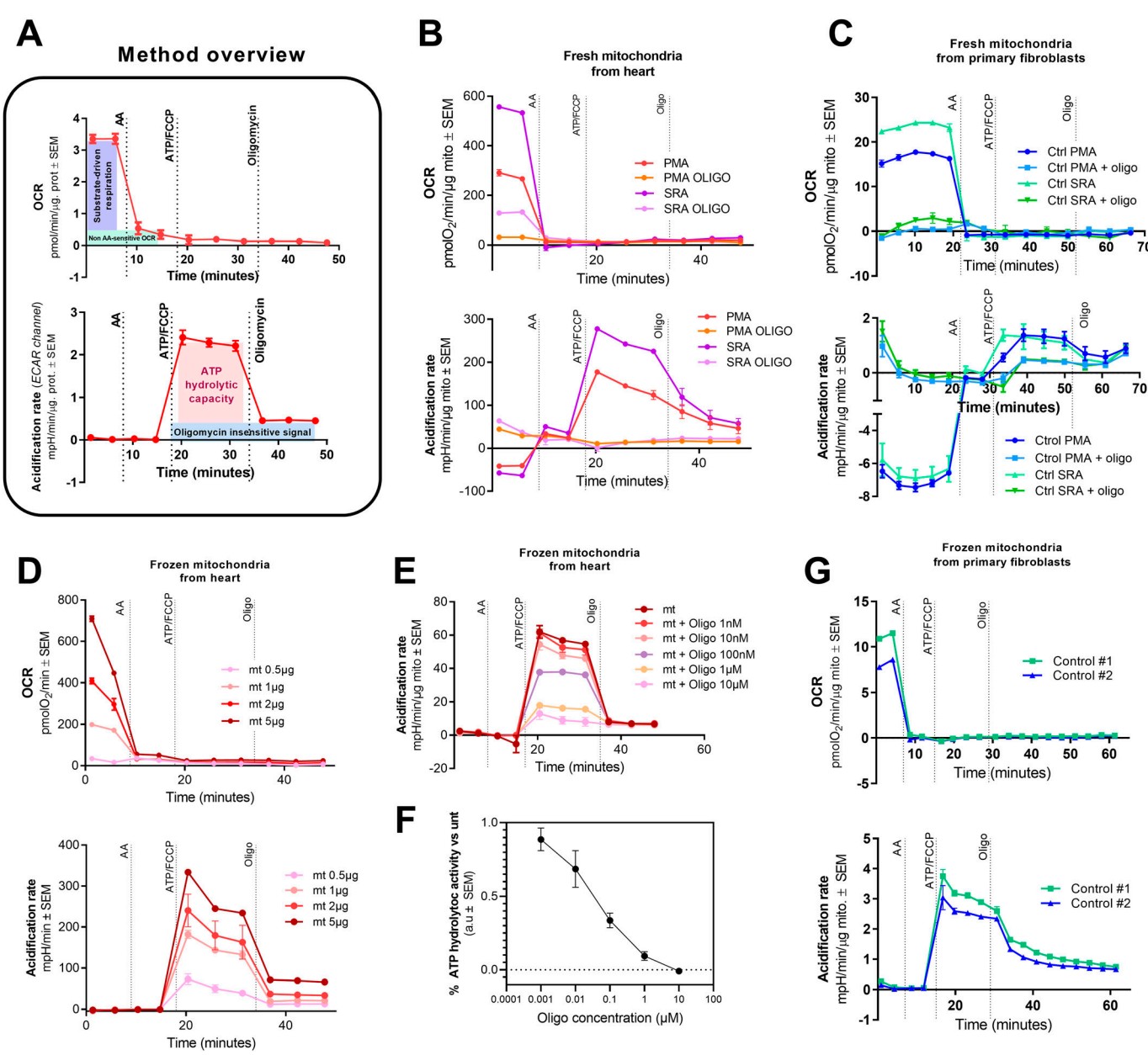

**Figure 2. Freezing helps with the detection of ATP hydrolytic capacity, especially in primary cells.**
**(A)** Overview of the ATP hydrolysis assay. Representative trace of oxygen consumption rate (OCR) (top) of a sample sustained with the substrates of interest. Representative acidification rate (ECAR channel) trace from the same assay (bottom). **(B)** Representative OCR traces of fresh mitochondria isolated from mouse heart sustained by pyruvate/malate/ADP or succinate/rotenone/ADP with and without oligomycin (top). Representative acidification rate traces from the same assay (bottom). **(C)** Representative OCR traces of fresh mitochondria isolated from primary human control fibroblasts sustained by pyruvate/malate/ADP or succinate/rotenone/ADP with and without oligomycin (top). Representative acidification rate traces from the same assay (bottom). **(D)** Representative OCR traces using different concentrations of previously frozen mitochondria isolated from mouse heart sustained by succinate/rotenone (top). Representative acidification rate traces from the same assay (bottom). **(E)** Representative acidification rate traces from previously frozen mitochondria isolated from mouse heart treated with the designated concentrations of oligomycin. Oligomycin concentrations depicted in the legend were added since the beginning of the assay. **(F)** Oligo-sensitivity of the ATP hydrolysis activity calculated from data shown in the previous panel. **(G)** Representative OCR traces of previously frozen mitochondria isolated from primary human control fibroblasts fueled by succinate/rotenone (top). Representative acidification rate traces from the same assay (bottom). #1 and #2 represent two biological replicates. In all cases, average ± SEM are shown. Source data are available for this figure.

from which we removed mitochondria. Cytosolic samples do not show any changes in the AR in response to ATP + FCCP (Fig 3B). Different tissues may need a different initial amount of starting material to achieve a good signal. For HyFS, doubling the amount of

protein used for the respirometry in frozen sample (RIFS) assay (Acin-Perez et al, 2020a) was sufficient sample mass to measure hydrolytic capacity for most of the samples tested (see a guideline for the concentration to use in different tissues in Table 1).

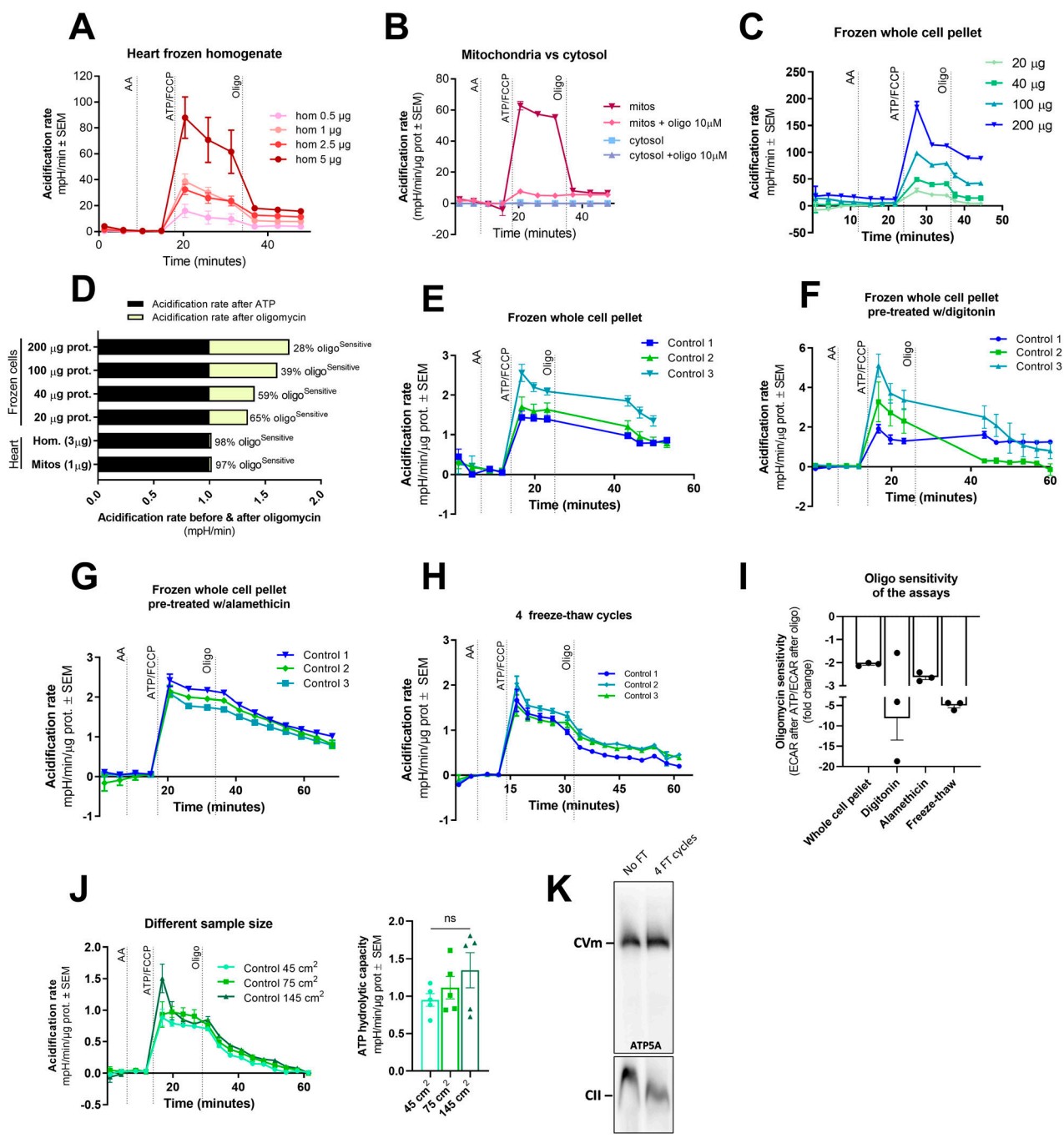

**Figure 3. Determination of ATP hydrolytic capacity is possible in frozen homogenates and whole-cell lysates.**
**(A)** Representative acidification rate traces from different concentrations of mouse heart homogenate. **(B)** Representative acidification rate traces from previously frozen mitochondrial and cytosolic fractions isolated from mouse heart, treated with and without oligomycin. **(C)** Representative acidification rate traces from different concentrations of previously frozen whole-cell lysates prepared from primary human fibroblasts. **(D)** Comparison of acidification rate values after the injection of ATP and after the injection of oligomycin from the traces shown in the previous panel, including a comparison with heart mitochondria and heart homogenate. **(E)** Representative acidification rate traces from 25 μg of previously frozen whole-cell lysates prepared from primary human fibroblasts. **(F, G, H)** Representative acidification rate traces from 25 μg of previously frozen whole-cell lysates prepared from primary human fibroblasts and pre-treated with digitonin (F), pre-treated with alamethicin (G), or exposed to four freeze–thaw cycles (H). **(I)** Oligomycin sensitivity comparison from the assays shown in panels (E, F, G, H). **(J)** Representative acidification rate traces from different size pellets of frozen whole-cell lysates prepared from primary human fibroblasts after four freeze–thaw cycles (left). ATP hydrolytic capacity quantification of the assay (right). **(K)** Blue native blots showing the assembly of CV in control samples (no FT) and in samples prepared after four freeze–thaw cycles (4 FT cycles). Control #1, #2, and #3 represent independent biological replicates. In all cases, average ± SEM are shown.
Source data are available for this figure.

**Table 1. Initial amount of protein recommended per tissue to perform ATP hydrolytic capacity determinations.**

| Tissue | RIFS | HyFS |
|---|---|---|
| Heart | 1–2 µg | 2.5–3 µg |
| Liver | 6–8 µg | 8–10 µg |
| Muscle | 6–8 µg | 20–25 µg |
| Kidney | 6–8 µg | 8–10 µg |
| Cell lysates | 8–12 µg | 20–25 µg |

Comparison with the amount used for RIFS.

After demonstrating that our method to determine ATP hydrolytic capacity works with tissue homogenates, we set our efforts into adapting the assay for previously frozen cell pellets obtained from primary human fibroblasts. Given that isolating mitochondria from cells requires a relatively large amount of starting material, we wanted to test whether cell pellet homogenates would decrease the required amount of starting material to run HyFS, which would make it a more feasible method for primary cells. In Fig 3C, we show that 200 µg of cell lysate obtained the highest increase in acidification. However, as we increase the amount of protein, the specificity of the assay decreases, as shown by the decrease in the ability of oligomycin to inhibit acidification when protein content was high, suggesting that under this condition either hydrolase other than CV becomes a significant contributor to the acidification signal or that the membranes are not sufficiently permeable for the ATP to get in the mitochondrial matrix (Fig 3C and D). Although the sensitivity to oligomycin was 98% in murine heart homogenates, it decreases to 28% when 200 µg of cell lysate was used in the assay (Fig 3D). We decided to keep the amount of protein relatively low (25 µg) and to extend the measuring time after the injection of oligomycin to allow for a better inhibition over time (Fig 3E). We have maintained the same amount of protein in our assays with cellular lysates henceforth. Yet, the inhibition of acidification by oligomycin was still very limited (Fig 3E), and further optimization was needed to assure that the assay is reliable and the quantification of ATP hydrolytic capacity by CV is accurate and reproducible.

We rationalized that the difference between the ability of oligomycin to inhibit acidification in cells versus tissue might not be due to different contents of hydrolases but rather due to the membranes having different resilience to mechanical permeabilization. With that in mind, we decided to test different permeabilization strategies in cell pellets to further break down membranes. First, we pre-treat the pellet with a low amount of digitonin, a strategy that is commonly used when measuring CV activity in gel (Diaz et al, 2009; Acin-Perez et al, 2020b). Digitonin increased the acidification signal, and in most cases, the inhibition by oligomycin was also enhanced (Fig 3F). However, the effect of this detergent was dramatically different across different samples (Fig 3F and I), likely as a result of the way digitonin permeabilizes the cells, which to a large extent varies by cell type, plasma membrane lipid content, and the initial number of cells. Therefore, digitonin was ruled out as a permeabilization strategy. Next, we tried to further permeabilize the cell pellets by including alamethicin in our assay. Alamethicin is a pore-forming peptide extensively used in permeabilized cell bioenergetic assays (Gostimskaya et al, 2003; Divakaruni et al, 2018a). Alamethicin addition was able to increase oligomycin sensitivity (Fig 3G), improving our determination of ATP hydrolytic capacity by CV in previously frozen cell lysates. As an alternative, we tested a mechanical approach consisting of four cycles of freezing and thawing the cell pellets (Fig 3H). When compared to the previously mentioned strategies, disruption by four freeze–thaw cycles was by far the most effective at boosting oligomycin sensitivity (Fig 3I), and it was also the most cost-effective because no detergents or permeabilizing agents were required. Henceforth, four freeze–thaw cycles were included in all HyFS protocols involving frozen cell lysates.

To test whether the size of the cell pellet (based on the number of cells being frozen) could modify ATP hydrolysis readout and its sensitivity to oligomycin, we grew cells on different culture plates and ran the assay using the freeze-thawing technique. We confirmed that the initial size of the cell pellet does not affect CV hydrolytic capacity determinations (Fig 3J). Finally, to confirm that the amount of fully assembled CV able to participate in ATP hydrolysis is not compromised by the freeze–thaw cycles, we performed a comparison using blue native gel electrophoresis (BNGE) analysis. We compared the native profile of assembled CV between a known protocol for native gel sample preparation (no freeze–thaw, no FT) and a preparation done after four freeze–thaw cycles (4 FT cycles) (Fig 3K). No differences were observed in the assembled CV monomer, confirming that freeze–thaw cycles do not compromise CV hydrolytic activity by compromising CV assembly in frozen samples. Note that CV dimer is not detectable in preparations coming from primary cells.

### Normalization of ATP hydrolytic capacity by CV

Changes in mitochondrial mass, mitochondrial activity, and mitochondrial protein expression can be revealed as changes in CV activity when measured in total tissue and cell lysates. Therefore, to determine whether there is a specific change in CV activity, there is the need to normalize CV activity by mitochondrial mass, respiratory activity, and/or total CV protein content or assembly. To illustrate how different normalization parameters can affect CV ATP hydrolytic capacity results, we compared ATP hydrolytic capacity per CV in two tissues with known differences in mitochondrial abundance per total protein: heart and liver.

First, the fluorescent dye MitoTracker Deep Red (MTDR) was used to estimate mitochondrial content in previously frozen homogenates. As expected, mitochondria are more abundant in heart homogenates when compared to liver homogenates (Fig 4A). As reported previously for the RIFS assay (Acin-Perez et al, 2020a; Osto et al, 2020), MTDR determinations can be done in parallel with the HyFS assay, by preparing a sister 96-well plate with the same samples. By incorporating this approach into the workflow of the HyFS assay, we can have an additional parameter to normalize the hydrolysis data. Normalizing ATP hydrolytic activity by the amount of cell or tissue protein loaded in the assay provides the differences in activity between two given samples. However, because ATP hydrolytic capacity is directly influenced by the mitochondrial content of a sample, we may use the MTDR determination to calculate hydrolytic capacity per mitochondrial mass. In our

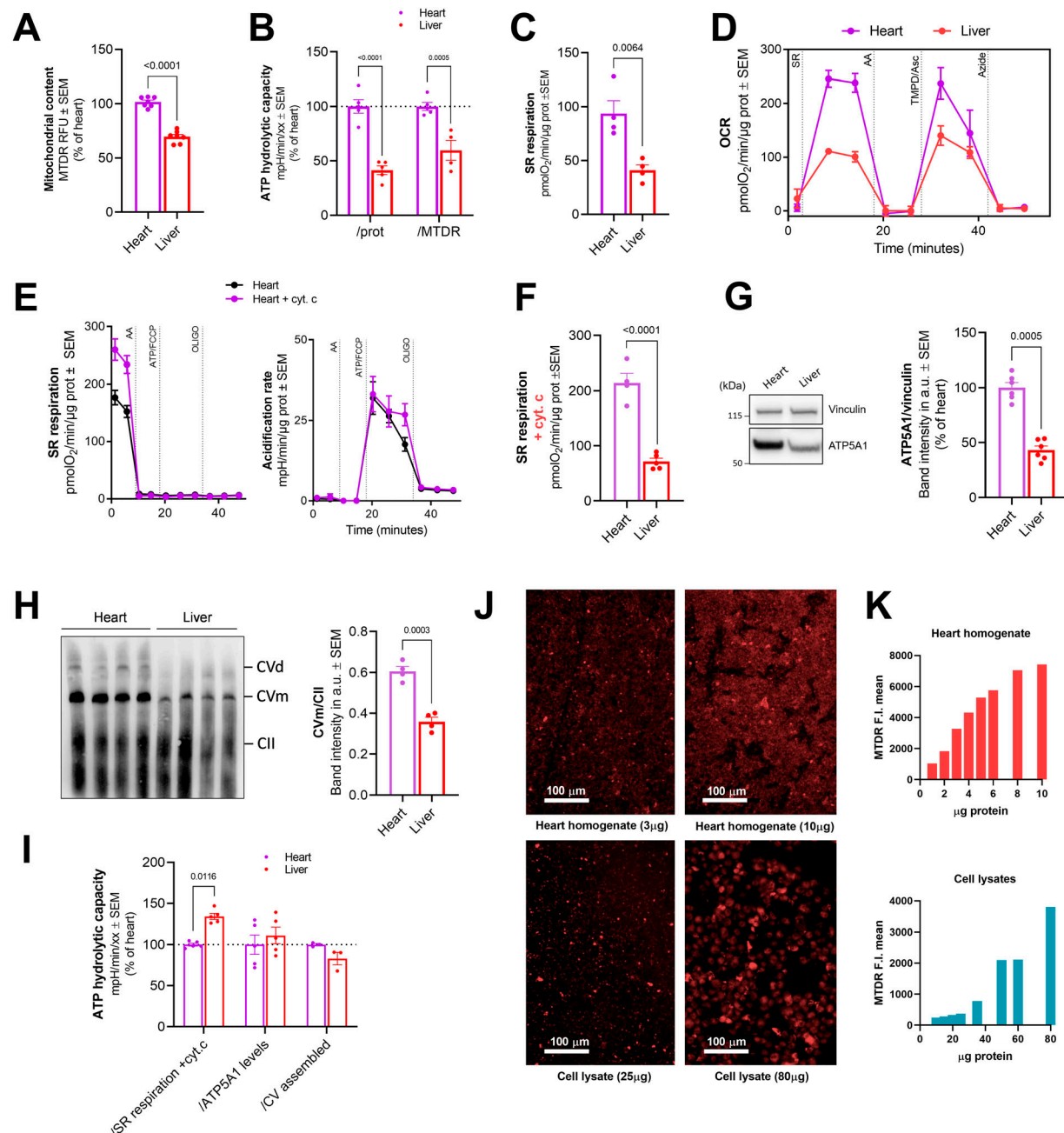

**Figure 4. Parameters to consider to normalize ATP hydrolytic capacity results.**
**(A)** Mitochondrial content measured by MTDR in frozen liver and heart homogenates (n ≥ 5). **(B)** ATP hydrolytic capacity normalized by protein or mitochondrial content in liver and heart homogenates (n ≥ 4). **(C)** SR respiration by μg of protein in frozen liver and heart homogenates (n = 4). **(D)** Representative succinate/rotenone OCR profile using the RIFS assay in frozen mouse heart and liver homogenates. **(E)** Representative succinate/rotenone-sustained OCR traces in mouse frozen heart homogenates plus and minus cytochrome c (left). Representative acidification rate traces from the same experiment (right). **(F)** SR respiration in frozen liver and heart homogenates from an assay containing cytochrome c (n ≥ 4). **(G)** Representative immunoblot bands (left) and the corresponding quantification (right) of ATP5A1 (n ≥ 6) from liver and heart homogenates. Vinculin is used as a loading control. **(H)** Blue native immunoblots (left) showing the assembled CV supercomplexes, both dimer (CVd) and monomer (CVm). CII assembly is used as a loading control. Quantification (right) of the mentioned immunoblots. **(I)** ATP hydrolytic capacity normalized by SR respiration + cyt. c., ATP5A1 levels, and CV assembled in frozen mouse heart and liver homogenates (n ≥ 3). **(J)** Representative micrographs of MTDR fluorescence intensity from the indicated concentrations of heart homogenate and cell lysates. Fluorescence was threshold for each image independently to avoid saturation. Scale bar = 100 μm. **(K)** Dose-dependent curves of MTDR in heart homogenates and cell lysates. Fluorescence intensity was measured by Operetta imager. Individual dots represent different biological replicates. Data shown are mean ± SEM.
Source data are available for this figure.

example, liver showed a lower ATP hydrolytic capacity than heart when normalizing per protein loaded (Fig 4B), and this difference persists after accounting for mitochondrial content differences (Fig 4B).

As described before, the HyFS assay starts with the determination of CII maximal respiration in an attempt to maximize the data obtained from limited samples. Different samples can differ not only in their mitochondrial amount but also in the activity of their mitochondria. Maximal SR respiration is a good readout of mitochondrial activity, and so it has the potential to become a useful normalizing parameter. We find that liver and heart homogenates differ when tested for SR respiration (Fig 4C) where heart homogenates show significantly higher rates. To ensure that we were accounting for the maximum activity of CII in our ATP hydrolytic capacity assay, we first identified the maximal CII and CIV activities in our homogenates using the RIFS assay (Acin-Perez et al, 2020a; Osto et al, 2020) (Fig 4D). We observed that the signal for SR respiration coming from the RIFS assay (Fig 4D) was higher than the values obtained in the HyFS assay (Fig 4C). This discrepancy indicates that the assay's design may have overlooked an additional element. A notable difference between the two assays is the presence of cytochrome c in the RIFS assay. Cytochrome c is added to the assay buffer to assure maximal activity because this soluble element of the electron transport chain is lost after freeze-thawing due to mitochondrial membrane permeabilization (Acin-Perez et al, 2020a). Therefore, to ensure that we are accurately measuring maximal respiration in our HyFS assay we decided to introduce cytochrome c in the assay buffer. The addition of cytochrome c increases SR respiration rates in our samples (Fig 4E, left, and Fig 4F) matching the rates measured by RIFS (Fig 4D), without affecting CV hydrolytic capacity (Fig 4E, right). Absolute values for maximal SR respiration are increased in liver and heart homogenates after the introduction of cytochrome c, but the differences observed between the two homogenates persist (Fig 4E and F). We used the readout of CII maximal OCR as an additional parameter to normalize ATP hydrolytic activity by CV. Doing so, we observed that hydrolytic capacity in liver is higher than in the heart (Fig 4H). This normalization becomes especially relevant for samples where other normalization parameters are unavailable because of technical difficulties or sample amount limitations.

Because ATP hydrolysis is a direct activity of CV, we could also opt for a more direct normalization approach. If sample size allows, protein levels of one of the proteins that form CV, ATP5A1, can be measured by Western blot and used to normalize CV ATP hydrolytic capacity per amount of one of the subunits of the enzyme responsible for the activity. In our example, ATP5A1 expression and thus approximated levels of ATP synthase are lower in liver when compared to heart homogenate (Fig 4G). For an even more accurate approach, the assembly of CV supercomplexes can be determined and used as a normalization parameter, because CV assembled is the enzyme responsible for the hydrolytic activity and not a single protein subunit. Native protein analysis shows that the amount of assembled CV (monomer and dimer) is higher in heart homogenates compared with liver (Fig 4H). As a result, when comparing liver and heart using ATP5A1 protein levels or the amount of CV assembled as a normalization parameter, we did not see any significant changes between the two homogenates (Fig 4I), meaning

that hydrolytic capacity of both tissues is correlated to the amount of CV present in each of these tissues.

Despite the fact that each normalization parameter has benefits and drawbacks associated with it, it is crucial to be aware of the various normalization factors that can be used to normalize ATP hydrolytic capacity. The sort of sample being examined, the justification for a particular project, and the exact question posed may all have an impact on the parameter to be used. Advantages and disadvantages of each normalization parameter described above are summarized in Table 2.

It is important to note here that the utilization of MTDR as a marker for mitochondrial mass can lead to artifacts when working with cell lysates. We have compared the staining of MTDR by imaging in heart homogenates and cell lysates at different protein concentrations (Fig 4J and K). Our images indicate that although in heart, the dye binds mainly to mitochondria (Fig 4J, top), in cell lysates, the dye is highly unspecific, binding to cellular debris or other cell membranes (Fig 4J, bottom). Contrary to the heart, where the signal increases linearly until it reaches saturation (Fig 4K, top), the unspecific binding of MTDR in cell lysates led to inaccurate and erratic quantifications that do not reflect mitochondrial mass (Fig 4J, bottom). We cannot discard a similar behavior in other types of samples with high amount of membranes and low content of mitochondria including brain homogenates. In those cases, the utilization of MTDR must be critically evaluated.

### Validation of ATP hydrolytic capacity assay per CV

To demonstrate the sensitivity of our method detecting changes in CV ATP hydrolytic capacity in primary cells, we examined a variety of models in which ATP hydrolysis is affected. First, we established a cell line stably expressing the constitutively active form of ATPIF1, the master regulator of ATP hydrolysis by CV (Gledhill et al, 2007). Primary control fibroblasts were used for this purpose (Fig 5A–D). An immunoblot showing the overexpression of the ATPIF1 active form in our cell model is shown in Fig 5C. This active form, named ATPIF1-H49K (Schnizer et al, 1996), selectively blocks the reverse rotation of CV and therefore compromises its ATP hydrolytic capacity. Previously frozen cell lysates obtained from control cells and cells overexpressing ATPIF1-H49K were compared using the HyFS assay. This comparison confirmed that ATP hydrolytic capacity per CV is decreased in cells overexpressing ATPIF1-H49K (Fig 5A). Neither SR respiration nor ATP5A1 levels change in cells overexpressing ATPIF1-H49K (Fig 5B and C); hence, the decrease in ATP hydrolytic capacity is maintained when these parameters are used for normalization (Fig 5D).

We also tested the opposite approach by knocking down *ATPIF1* in HEK293T cells. In this approach, two independent shRNAs reduced the expression of ATPIF1 to 50% or less when compared to the expression of cells transfected with a scrambled control shRNA (Fig 5E). When the ATP hydrolytic capacity was analyzed in these samples, we observed that decreasing the amount of ATPIF1 available to block the rotation of CV impacts its capacity to hydrolyze ATP, when the hydrolytic activity is higher than in control samples (Fig 5F). Neither SR respiration (Fig 5G) nor ATP5A1 levels (Fig 5E) were altered in cells with decreased ATPIF1 levels; therefore,

**Table 2. Advantages and disadvantages of the different parameters that can be used to normalize ATP hydrolytic capacity.**

| Normalizing parameter | Rationale | Advantages | Disadvantages |
|---|---|---|---|
| Mitochondrial content (by MitoTracker Deep Red) | Normalize ATP hydrolytic capacity measurements by the mitochondrial content present in each sample, which can vary. | Can be done in parallel with the ATP hydrolytic capacity assay using the same sample dilutions. | In cell lysates, the number of mitochondria is low and the dye is highly unspecific. It won't represent mitochondrial content. |
| Succinate/rotenone respiration | Normalize ATP hydrolytic capacity measurements by the OXPHOS activity of each sample, represented by the maximal respiratory activity of CII. | It is measured together with the ATP hydrolytic capacity working as a surrogate for mitochondrial content when other determinations are not feasible. | If OXPHOS or CII respiration is significantly up- or down-regulated in a sample, the results can be misleading. |
| ATP5A1 expression | Normalize ATP hydrolytic capacity measurements by the expression of one of the constitutive proteins of CV, which is directly accountable for the activity. | Correlate ATP hydrolysis activity directly to the approximate amount of enzyme responsible for the activity. | Because immunoblots are required, it can make the study more complicated. In cell lysates, sample is often insufficient and Western blots are usually performed on a separate set of samples. Combining data from different techniques usually increases the error. |
| Assembled CV | Normalize ATP hydrolytic capacity measurements by the amount of assembled CV, which is directly accountable for the activity. | Correlate ATP hydrolysis activity directly to the amount of enzyme responsible for the activity. | Because native protein immunoblots are required, it makes the analysis more laborious. In cell lysates, it is challenging to achieve the amount of sample needed and it will always be performed on a separate set of samples. In addition, native blots include the use of detergents such as digitonin to extract the complexes from the membranes, which will involve extra optimization steps and will add an extra layer of complexity and error to the results. |

the increase in ATP hydrolytic capacity is maintained when these parameters are used for normalization (Fig 5H).

Finally, as an additional approach to demonstrate that our method can detect physiological changes in CV hydrolytic capacity, we tested the developed assay after pharmacological interventions that force CV in cells into a hydrolytic mode. We treated primary control fibroblasts with low concentrations of AA and FCCP for 24 and 18 h, respectively. Then, we collected the cell lysate pellets for examination. Either by the blockage of respiration (AA) or by the dissipation of the membrane potential (FCCP), both drugs force CV to hydrolyze ATP at a higher-than-normal rate to keep the cells alive. After running the HyFS assay, we observed that control cells incubated with both 100 nM AA and 1 $\mu$M FCCP increase ATP hydrolytic capacity (Fig 5I). These drugs have a dramatic impact on respiration as observed in Fig 5J. Cells treated with AA show more than 80% reduction in SR respiration, whereas cells treated with FCCP have around 50% increase in respiration (Fig 5J). These drastic differences exemplify the importance of considering different types of normalization based on the type of sample being studied. In this specific scenario, where mitochondrial activity is so highly impacted, normalizing by SR respiration (Fig 5L) can lead to misleading results as stated in Table 2, and should be avoided. Immunoblots against ATP5A1 reveal that our intervention with FCCP or AA has no effect on the expression of this protein (Fig 5K), confirming the increase in ATP hydrolytic capacity when CV content is used as a normalizer (Fig 5L).

Overall, we demonstrated that when ATP hydrolysis is altered because of different circumstances our method is able to detect these changes in previously frozen homogenates and cell lysates. Finally, to show that our method could be potentially used to detect ATP hydrolytic capacity per CV as a disease biomarker, we measured CV hydrolytic capacity in PBMCs (Fig 5M), a cell sample that can be easily collected and is routinely available from patients and that could be frozen for further analysis in clinical settings. This result, together with the discovery of the contribution of ATP hydrolysis in mitochondrial deficiencies (paper under review in EMBO J, EMBOJ-2022-111699), opens the possibility of assessing CV hydrolytic activity as a biomarker or contributor in mitochondrial deficiency disorders.

# Discussion

At the cellular level, ATP serves as both an energy source and an energy reservoir. Thus, it is very important that cells keep a constant level of ATP despite fluctuations in supply and demand. Abnormal regulation of ATP levels and ATP flux is a common denominator in cancer, cardiovascular diseases, obesity, neuronal disorders, and immune dysfunction among other disorders (Koopman et al, 2012; Ganapathy-Kanniappan & Geschwind, 2013; Pathak et al, 2013: Ganapathy-Kanniappan, 2013 #28, Cole, 2016; Johnson et al, 2019; Dunn & Grider, 2022), with mitochondria playing a key role in this

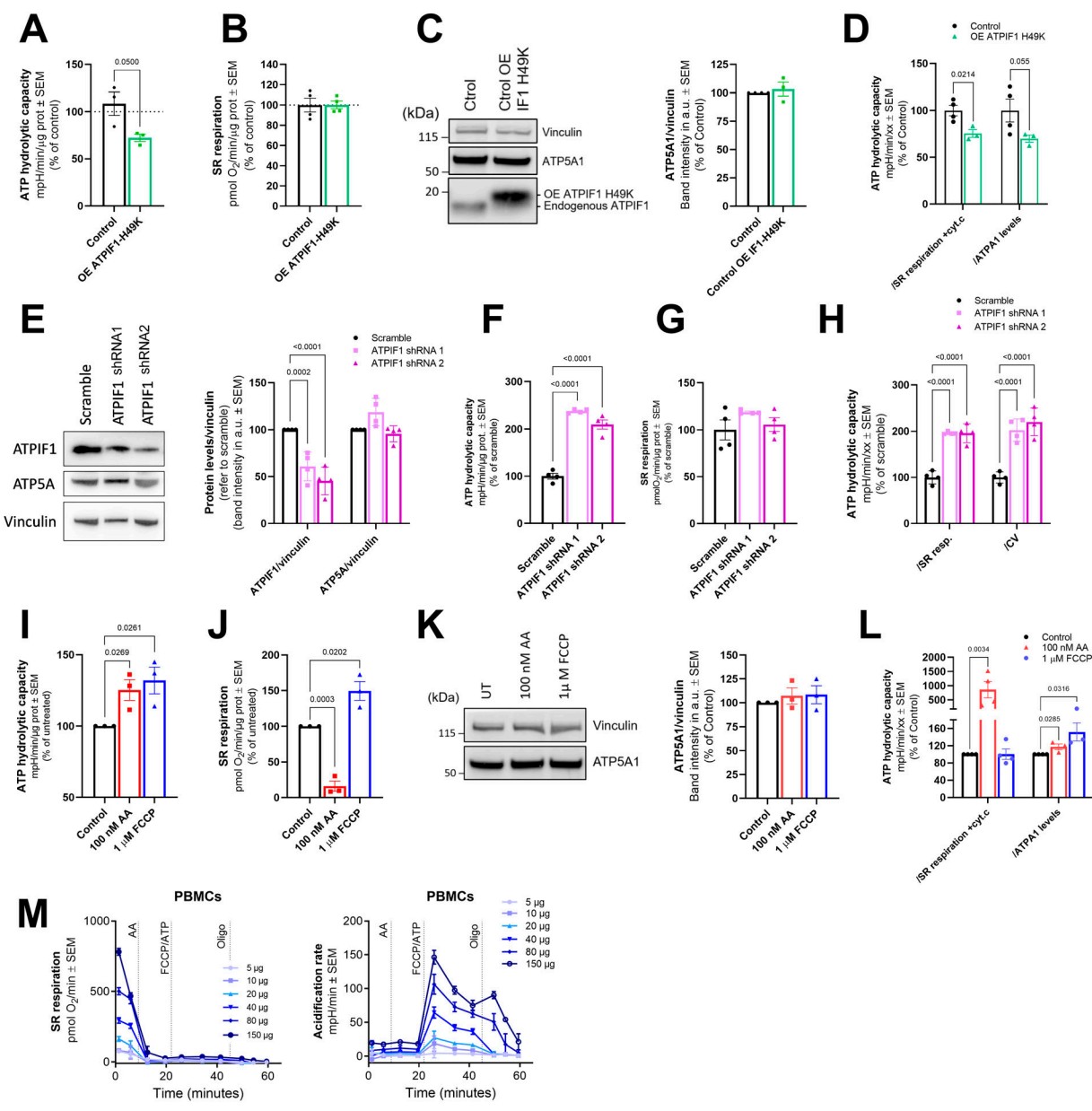

**Figure 5. Validation of the method using hydrolysis-inducing drugs and genetic strategies.**
**(A)** ATP hydrolytic capacity from frozen lysates of control cells and control cells overexpressing ATPIF1-H49K (n = 3). **(B)** SR respiration from frozen lysates of control cells and control cells overexpressing ATPIF1-H49K (n = 4). **(C)** Representative immunoblot bands (left) and the corresponding quantification (right) of ATP5A1 in control cells and control cells overexpressing ATPIF1-H49K (n = 4). Vinculin is used as a loading control. Immunoblot confirming the overexpression of ATPIF1-H49K in our model is also shown. **(D)** ATP hydrolytic capacity normalized by SR respiration + cyt. c. and ATP5A levels from control cells and control cells overexpressing ATPIF1-H49K (n = 4). **(E)** Representative immunoblots (left) and the corresponding quantification (right) of ATPIF1 and ATP5A1 in HEK293T cells transfected with a scramble control or two independent shRNAs against ATPIF1 (n = 4). Vinculin is used as a loading control. **(F)** ATP hydrolytic capacity from frozen lysates of HEK293T cells transfected with a scramble control or two independent shRNAs against ATPIF1 (n = 4). **(G)** SR respiration of the samples shown in (F). **(H)** ATP hydrolytic capacity normalized by SR respiration + cyt. c. and ATP5A levels from frozen lysates of HEK293T cells transfected with a scramble control or two independent shRNAs against ATPIF1 (n = 4). **(I)** ATP hydrolytic capacity from control human primary fibroblasts pre-treated with FCCP (18 h) and AA (24 h) at the stated concentrations (n = 3). **(J)** SR respiration in frozen lysates from control human primary fibroblasts pre-treated with FCCP and AA at the stated concentrations (n = 3). **(K)** Representative immunoblot bands (left) and the corresponding quantification (right) of ATP5A1 in control cells and control cells pre-treated with FCCP and AA (n = 3). Vinculin is used as a loading control. **(L)** ATP hydrolytic capacity normalized by SR respiration + cyt. c. and ATP5A1 levels from control cells and control cells pre-treated with FCCP and AA (n ≥ 3). **(M)** Representative OCR traces from previously frozen PBMCs fueled by succinate/rotenone (left). Representative acidification rate traces from the same assay (right). Data shown are mean ± SEM.
Source data are available for this figure.

process. Current studies on mitochondrial implications in ATP dynamics and homeostasis are mainly focused on assessing ATP synthesis by CV, measurements of total ATP levels (both mitochondrial and cytosolic), and the glycolytic switch. Other cellular events, such as the ATP hydrolytic activity of CV, have been also reported to contribute to ATP depletion (St-Pierre et al, 2000; Nieminen, 2003; Chinopoulos & Adam-Vizi, 2010). In fact, we recently demonstrated that high ATP hydrolysis by CV contributes to

pathogenesis in models of mitochondrial deficiencies (paper under review in EMBO J, EMBOJ-2022-111699). However, ATP hydrolysis by CV is not frequently studied because a direct, standardized, and accessible method to measure ATP hydrolysis is currently lacking.

Current approaches to measure ATP hydrolytic activity often involve indirect reactions, typically linked to the production of phosphorus or NADH (Sumner, 1944; Pullman et al, 1960; Gomez-Puyou et al, 1983; Galber et al, 2021). This is the case in the ATP-regenerating system that uses NADH absorbance as a readout, where the ATP added reacts with the supplemented pyruvate kinase, lactate dehydrogenase, and phosphoenolpyruvate in the assay buffer (Galber et al, 2021). The approach including the measurement of phosphorus is a colorimetric method that uses the phosphorus generated from ATP hydrolysis in a chemical reaction involving ammonium molybdate, sulfuric acid, and ferrous sulfate (Sumner, 1944). ATP hydrolytic activity can also be assessed in situ in native gels, where mitochondrial complexes remain assembled and active, by using in-gel activity methods specific for CV (Acin-Perez et al, 2020b). However, in-gel activity determinations are very laborious, require the use of detergents that may potentially affect the final activity reported, and demand isolated mitochondria. The need for isolated mitochondria is a common limitation among the multiple techniques described to measure ATP hydrolysis, with the exception of an indirect approach that used tissue homogenates (Haraux & Lombes, 2019) and our previous determination of ATP hydrolytic activity in HepG2-permeabilized cells (Divakaruni et al, 2018a). Although the mentioned methodologies are successful in particular scenarios, their major limitation is the use of detergents (Acin-Perez et al, 2008; Haraux & Lombes, 2019), alternative disruption procedures (Rustin et al, 1994; Mayr et al, 2004; Benit et al, 2006), and the amount of sample needed, which make them impractical for certain cellular cultures, clinical samples, and high-throughput studies.

Another key consideration is whether the CV measured activity is reflecting the synthetic or hydrolytic capacity of the complex. We need to be aware that depending on the nature of the sample and how the samples have been treated, we are often only measuring either synthesis or hydrolysis. It is common in the literature to measure CV enzymatic activity as a surrogate for ATP synthesis; however, these readouts are mostly addressing ATP hydrolysis. Knowing that ATP hydrolysis by CV can contribute to diseases independently of ATP synthesis, it is crucial to differentiate which activity is being measured in a particular assay. ATP synthesis activity or ATP synthase working in the forward mode can be measured directly in isolated mitochondria or permeabilized cells as described (Vives-Bauza et al, 2007) or indirectly by assessing state 3 respiration in the presence of substrates and ADP (Jones et al, 2021). On the contrary, it must be highlighted that methods that refer to maximal CV activity primarily evaluate ATP hydrolytic capacity.

Here, we describe a new methodology that allows the determination of maximal ATP hydrolytic capacity by CV and maximal respiratory activity driven by electron entry through complex II (or I) in the same assay. Our method provides a direct readout of ATP hydrolytic activity by monitoring changes in pH, eliminates the use of detergents or other disruptive chemicals, and is optimized to be used in previously frozen cell lysates and tissue homogenates,

without the requirement of isolating mitochondria. Overall, our approach greatly eases sample preparation while permitting a reasonable level of high-throughput research.

To ensure that the HyFS method can be easily extrapolated to any cell line of interest, our efforts were focused on optimization of ATP hydrolytic capacity determinations in primary skin biopsy–derived fibroblasts, a challenging cell type that has not been contemplated in any of the existing ATP hydrolysis methodologies. In mitochondrial diseases, skin biopsy–derived fibroblasts are one of the most commonly accepted models used to better understand the mechanisms underlying the disease (Hu et al, 2019; Acin-Perez et al, 2021). These cells are directly isolated from human biological samples preserving the morphological and functional features (Schneider, 1979; Auburger et al, 2012). However, using mutant primary derived fibroblasts can be challenging because they can grow very slowly depending on the mutation and have a limited number of passages before entering senescence (Campisi, 1996; Cristofalo et al, 2004; Schauble et al, 2012; Hernandez-Segura et al, 2018). For that reason, minimizing the required amount of sample, avoiding the need to isolate mitochondria, and simplifying the sample processing are required to perform bioenergetic studies in these cells. Our optimization includes multiple freeze-thawing cycles after freezing, which leads to CV accessibility through the total breakage of the inner membrane without the need for permeabilizing reagents and, more importantly, avoids the need for mitochondrial isolation.

In addition to fibroblasts, the described method can also be used with other type of samples. Indeed, we have shown that the HyFS method is compatible with previously frozen tissue homogenates and isolated mitochondria, making it highly versatile. In addition, we showed that our methodology can be successfully applied to human blood cells, PBMCs, opening the possibility to study the role of mitochondrial ATP depletion in large population-based health research using a non-invasive and clinically relevant sample.

Besides the advantages regarding the type of samples used in the analysis and the absence of detergents, the HyFS method opens the possibility to analyze ATP hydrolytic capacity by CV in a reasonably high-throughput procedure. The utilization of the 96-well format Seahorse platform allows for a less initial amount of biological material, whereas multiple technical replicas or independent biological samples can be measured at the same time. Also, as previously stated, our methodology allows for the determination of hydrolytic capacity and maximal respiration in the same assay. This dual result not only maximizes the information obtained from a single sample, but also contributes to result interpretation based on the samples used, as shown when SR respiration is altered in AA-treated cells, but ATP hydrolysis is increased (Fig 5K–N).

Three approaches were used to validate our methodology: (1) the overexpression of the always active CV hydrolytic activity inhibitor ATPIF1-H49K, which inhibits CV hydrolytic capacity (Fig 5A–D); (2) the partial silencing of ATPIF1 using shRNA, which enhances CV hydrolytic activity (Fig 5G–J); and (3) the pre-treatment of cells in culture with low concentrations of AA and FCCP, which increase CV hydrolytic capacity in the cell lysates (Fig 5K–N). Our experiments showed that different strategies are able to influence the hydrolytic activity of CV and that these alterations perdure over time, even after several freeze–thaw cycles, and are detectable when maximal

ATP hydrolytic capacity is measured with our methodology. However, the HyFS method also has some caveats. The change in hydrolytic capacity needs to be preserved with freeze–thaw, homogenization, or permeabilization; therefore, transient modifications of the activity may be lost during the assay resulting in erroneous negative results. Even when changes in ATP hydrolytic capacity are detectable, it is important to interpret the results correctly to avoid artifacts. For this purpose, we have established different normalization considerations and discussed their advantages and limitations (Fig 4 and Table 2).

Overall, our method to determine the maximal ATP hydrolytic capacity per CV can increase access to study ATP hydrolysis in diseases through retrospective research studies, clinical trials, and population exposure monitoring with the potential for new discoveries in science and medicine.

# Materials and Methods

### Tissue homogenization

Liver and heart from WT mice were homogenized in 0.5 ml MAS (70 mM sucrose, 220 mM mannitol, 5 mM $KH_2PO_4$, 5 mM $MgCl_2$, 1 mM EGTA, and 2 mM Hepes, pH 7.2) using 3-mm zirconium beads in a Bead Blaster (Benchmark). Two cycles at a speed of 6 m/s of 30 s on–30 s off were used. The homogenate was centrifuged at 1,530$g$ and the supernatant stored at –80°C.

### Cell culture

Primary fibroblasts (Beninca et al, 2021) derived from a healthy subject (control) were maintained in EMEM (Eagle's Minimum Essential Medium; # 30-2003; ATCC) supplemented with 15% FBS and antibiotic–antimycotic (#15240062; Thermo Fisher Scientific) in a humidified atmosphere at 37°C and 5% $CO_2$. 24 h before cell harvest, media were changed to phenol red–free DMEM (#A1443001; Thermo Fisher Scientific) containing 4.5 g/l glucose supplemented with 2 mM glutamine, 1 mM sodium pyruvate, 10% FBS, and antibiotic–antimycotic. Lentiviral infection of pLentiATPIF1-H49K (produced by Welgen Inc.) was performed using 8 $\mu$g/ml polybrene in EMEM for 30 h followed by a positive selection with 1 $\mu$g/ml puromycin. For drug interventions, AA and (carbonyl cyanide-p-trifluoromethoxyphenylhydrazone) FCCP were directly added to the growth media at the specified concentration.

The *ATPIF1* gene was knocked down by shRNA in HEK293T cells (ATCC) to facilitate transfection. Lipofectamine RNAiMAX (Thermo Fisher Scientific), Opti-MEM (Thermo Fisher Scientific), and two independent shRNAs against *ATPIF1* (#SASI_Hs01_00178339 and #SASI_Hs01_00230332; Sigma-Aldrich) were used, following manufacturer's instructions for a modified reverse transfection. Cells were harvested 48 h after transfection.

### Mitochondrial isolation

#### From heart
Mice were anesthetized with isoflurane and euthanized by cervical dislocation. The heart was removed and instantly placed in ice-cold relaxation buffer (5 mM sodium pyrophosphate, 100 mM KCl, 5 mM EGTA, and 5 mM Hepes, pH 7.4). The heart was squeezed with tweezers to remove blood, minced with scissors, and then placed in a glass–glass Dounce homogenizer with 3 ml of HES homogenization buffer (250 mM sucrose, 5 mM Hepes, and 1 mM EDTA, pH 7.2, adjusted with KOH). The heart tissue was homogenized with 10 strokes with the loose pestle and 15 strokes with the tight pestle. The homogenized tissue was placed in a pre-chilled 15-ml conical tube and centrifuged at 900$g$ (4°C) for 10 min. The supernatant was removed, placed in a new tube, and centrifuged again at 900$g$ for 10 min. The supernatant was then transferred to 2-ml microcentrifuge tubes and centrifuged at 10,000$g$ (4°C) for 10 min. The mitochondrial pellet was resuspended in ice-cold HES buffer, and the protein concentration was measured using a BCA assay (Pierce). After performing the assays with fresh mitochondria, the rest of the sample was stored at –80°C until further use.

#### From primary fibroblasts
Primary control fibroblasts from four confluent 150-cm$^2$ dishes were washed and harvested by trypsinization. Pellet was washed twice, resuspended in 1 ml of HES homogenization buffer, and placed in a glass–Teflon Dounce homogenizer. Cells were homogenized with 45 strokes and centrifuged at 600$g$. The supernatant was stored in a pre-chilled Eppendorf tube, and the pellet was rehomogenized and spun again to maximize the recovery. A combination of both supernatants was centrifuged at 9,000$g$ to collect the mitochondrial fraction. Mitochondrial pellet was finally resuspended in 30 $\mu$l of HES. Protein concentration was measured by BCA assay, experiments in fresh mitochondria were performed right away, and the rest of the sample was frozen at –80°C.

### HyFS assay

ATP hydrolytic capacity per CV determination was inspired by previous methodologies developed by us and others (McQuaker et al, 2013; Divakaruni et al, 2018a; Acin-Perez et al, 2021). The current assay also includes the determination of maximal respiratory capacity. HyFS assay was started as follows.

#### Isolated mitochondria
Heart (0.5–5 $\mu$g) or primary fibroblast (10–20 $\mu$g) mitochondria were loaded into a Seahorse XF96 microplate in 20 $\mu$l of MAS (70 mM sucrose, 220 mM mannitol, 5 mM $KH_2PO_4$, 5 mM $MgCl_2$, 1 mM EGTA, and 2 mM Hepes, pH 7.2) plus 1% free fatty acid/BSA containing substrates. The plate was centrifuged at 2,000$g$ for 5 min at 4°C (no brake), and then, 130 $\mu$l of MAS was added to each well. We started mitochondrial respiration in state 3 using the following substrates: (i) 5 mM pyruvate + 5 mM malate + 4 mM ADP or (ii) 5 mM succinate + 2 $\mu$M rotenone + 4 mM ADP. Under conditions where oligomycin was added at the beginning of the assay, a concentration of 1 $\mu$M oligomycin was used.

#### Tissue homogenates
Heart (0.5–5 $\mu$g) or liver (10 $\mu$g) homogenates were loaded into a Seahorse XF96 microplate in 20 $\mu$l of MAS. The plate was centrifuged at 2,000$g$ for 5 min at 4°C (no brake), and then, 130 $\mu$l of MAS containing 5 mM succinate/2 $\mu$M rotenone was added to each well.

From Fig 4E onward, 100 μg/ml of cytochrome c was also added to the 130 μl of MAS.

### Cell lysates

Frozen cell pellets coming from a 10-cm$^2$ dish were (i) resuspended in 100 μl of MAS buffer and used in the assay, (ii) solubilized using 50 μl of an 8 mg/ml digitonin stock, or (iii) put through four freeze–thaw cycles alternating between liquid nitrogen and a 37°C water bath. When a digitonin treatment was performed, the pellet was incubated for 10 min on ice, centrifuged for 10 min at 10,000$g$, and then resuspended in 30 μl MAS. Protein concentration was measured, and 20–200 μg of cell lysate was loaded into a Seahorse XF96 microplate in 20 μl of MAS. The plate was centrifuged at 2,000$g$ for 5 min at 4°C (no brake), and then 130 μl of MAS containing 5 mM succinate/2 μM rotenone was added to each well. From Fig 4E onward, 100 μg/ml of cytochrome c was also added to the 130 μl of MAS. When alamethicin permeabilization was used, a final concentration of 10 μg/ml of the compound was added to the final 130 μl of MAS.

### PBMCs

PBMC samples were centrifuged at 300$g$ for 10 min. The supernatant was discarded and the pellet put through four freeze–thaw cycles. 5–150 μg of sample was loaded into a Seahorse XF96 microplate in 20 μl of MAS. The plate was centrifuged at 2,000$g$ for 5 min at 4°C (no brake), and then 130 μl of MAS containing 5 mM succinate/2 μM rotenone and 100 μg/ml of cytochrome c was added to each well.

Then, in all cases, the Seahorse method was initiated and injections were performed as indicated in the figure panels at the following final concentrations: AA (2 μM), FCCP (1 μM), ATP (20 mM), and oligomycin (5 μM). To achieve maximal ATP concentration, two consecutive injections of ATP/FCCP were performed.

### RIFS assay

Respiration in previously frozen samples was performed as previously described (Acin-Perez et al, 2020a; Osto et al, 2020) in MAS containing 100 μg/ml of cytochrome c.

### Mitochondrial content

To determine mitochondrial mass in tissue homogenates, 5 μg of heart or 10 μg of liver in 20 μl of MAS was placed in a clear-bottom black 96-well microplate. Then, 130 μl of a 1:2,000 dilution of the 1 mM MTDR FM stock (MTDR; Thermo Fisher Scientific) was added and incubated for 10 min at 37°C. The plate was centrifuged at 2,000$g$ for 5 min at 4°C (without brake), and the supernatant was carefully removed. Finally, 100 μl of PBS was added per well and MTDR fluorescence measured ($\lambda$excitation = 625 nm; $\lambda$emission = 670 nm). Mitochondrial content was calculated as MTDR signal (minus blank) per microgram of protein. Homogenates and cell lysates were also imaged after the addition of MTDR with a PerkinElmer Operetta CLS high-content system (20× objective). Analysis was performed with PerkinElmer Harmony software (Harmony 4.1) by measuring mean MTDR fluorescence intensity in the whole imaged field.

## Protein gel electrophoresis and immunoblotting

### SDS–PAGE

A confluent 10-cm$^2$ dish per cell line was washed and lysed using RIPA buffer (0.3 M NaCl, 0.1% SDS, 50 mM Tris, pH 7.4, 0.5% deoxycholate, 1 mM Na$_3$VO$_4$, 10 mM NaF, 10 mM MgCl$_2$, and 1% n-dodecyl-$\beta$-D-maltoside, with phosphatase and protease inhibitors). 15–30 μg of tissue homogenate or RIPA cell lysate was loaded into 4–12% Bis-Tris gels (Thermo Fisher Scientific), and gel electrophoresis was performed in xCell SureLock Mini-Cells (Thermo Fisher Scientific) under a constant voltage of 120 V.

### BNGE

Tissue lysates (100–150 μg) were centrifuged at 10,000$g$ for 10 min. The pellet containing mitochondria was resuspended in 30–50 l of MAS, and protein amount was determined by BCA. Digitonin (DIG) incubation (liver, 3 mg DIG/mg protein; heart, 6 mg DIG/mg protein) was performed on ice for 5 min and then centrifuged at 20,000$g$ for 30 min as previously described (Acin-Perez et al, 2008; Acin-Perez et al, 2020a). Cell preparations for BNGE were performed as described (Fernandez-Vizarra & Zeviani, 2021). Supernatants containing mitochondrial complexes and supercomplexes were mixed with blue native sample buffer (5% Blue G dye in 1 M 6-aminohexanoic acid), loaded, and run on a 3–12% native precast gel (Invitrogen). Gels were run till the blue front ran out of the gel and the gel was transferred to PVDF membranes.

### Immunoblotting

Proteins were transferred to a methanol-activated PVDF membrane in a Mini Trans-Blot cell (Bio-Rad) at a constant voltage of 100 V for 75 min on ice. Blots were blocked in 3% BSA in PBS–Tween-20 (1 ml/l) for 1 h and incubated with the primary antibody overnight. Primary antibodies used were ATP5A1 (Invitrogen), ATPIF1 (Cell Signaling Technology), SDHA (Invitrogen), and vinculin (Sigma-Aldrich). The next day, membranes were washed three times with PBS-T, incubated with the adequate HRP-conjugated secondary antibody, and washed three more times with PBS-T. Images were acquired in a ChemiDoc Imaging System (Bio-Rad), and band densitometry was quantified using Image Lab (Bio-Rad).

## Statistical analysis

Statistical analyses were performed using GraphPad Prism 9.01. All the data shown are the mean ± SEM from at least three biological replicates. Means were compared using either a two-tailed $t$ test or one-way ANOVA. Individual points in a graph denote different biological replicates. Differences were considered statistically significant at $P < 0.05$, and the exact $P$-value is annotated in the different panels.

# Supplementary Information

# Acknowledgements

The authors would like to acknowledge Dani Dagan for critical reading of the article and valuable discussion. This work was supported by National Institutes of Health grants: CURE—DDRC NIH-NIDDK, P30 DK041301 (M Liesa); UCLA/UCSD DERC NIH-NIDDK—P30 DK063491 (M Liesa), R01 DK099618-05 (OS Shirihai), R01 CA232056-01 (OS Shirihai), R21AG060456-01 (OS Shirihai), and R21 AG063373-01 (OS Shirihai); ADA—1-19-IBS-049 (OS Shirihai) and R35GM138003 (AS Divakaruni); the WM Keck Foundation (AS Divakaruni); and Seed Award from DGSOM at UCLA (M Liesa).

## Author Contributions

L Fernandez-del-Rio: conceptualization, data curation, formal analysis, validation, investigation, visualization, methodology, and writing—original draft, review, and editing.
C Benincá: conceptualization, resources, data curation, formal analysis, supervision, validation, investigation, visualization, methodology, and writing—original draft, review, and editing.
F Villalobos: validation and investigation.
C Shu: validation, investigation, and methodology.
L Stiles: conceptualization, resources, supervision, funding acquisition, validation, investigation, visualization, methodology, and writing—review and editing.
M Liesa: conceptualization, resources, supervision, funding acquisition, and writing—review and editing.
AS Divakaruni: conceptualization, methodology, and writing—review and editing.
R Acin-Perez: conceptualization, data curation, formal analysis, supervision, validation, investigation, visualization, methodology, and writing—original draft, review, and editing.
OS Shirihai: conceptualization, resources, supervision, funding acquisition, visualization, and writing—review and editing.

## Conflict of Interest Statement

OS Shirihai is a co-founder and SAB member of Enspire-Bio and Capacity-Bio, and has been serving as a consultant to LUCA-Science, IMEL, Epirium, Johnson & Johnson, Pfizer, and Stealth Biotherapeutics.

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
