## [Reviewer comments · Life Science Alliance]

Life Science Alliance

A Novel Approach to Measure Complex V ATP Hydrolysis in Frozen Cell Lysates and Tissue Homogenates

Orian Shirihai, Lucia Fernandez-del-Rio, Cristiane Benincá, Frankie Villalobos, Cynthia Shu, Linsey Stiles, Marc Liesa, Ajit Divakaruni, and Rebeca Acín-Pérez

DOI: <https://doi.org/10.26508/lsa.202201628>

Corresponding author(s): Orian Shirihai, University of California, Los Angeles and Rebeca Acín-Pérez

Review Timeline:

Submission Date:	2022-07-25
Editorial Decision:	2022-09-02
Revision Received:	2022-10-27
Editorial Decision:	2022-11-22
Revision Received:	2022-11-30
Accepted:	2022-12-01

Scientific Editor: Novella Guidi

Transaction Report:

September 2, 2022

Re: Life Science Alliance manuscript #LSA-2022-01628

Prof. Orian S. Shirihai
University of California, Los Angeles
Department of Molecular and Medical Pharmacology, David Geffen School of Medicine
650 Charles East Young Drive South
CHS 27-200
Los Angeles, CA 90095

Dear Dr. Shirihai,

Thank you for submitting your manuscript entitled "A Novel Approach to Measure Complex V ATP Hydrolysis in Frozen Cell Lysates and Tissue Homogenates" to Life Science Alliance. The manuscript was assessed by expert reviewers, whose comments are appended to this letter. We invite you to submit a revised manuscript addressing the Reviewer comments.

Thank you for this interesting contribution to Life Science Alliance. We are looking forward to receiving your revised manuscript.

Sincerely,

B. MANUSCRIPT ORGANIZATION AND FORMATTING:

Reviewer #1 (Comments to the Authors (Required)):

This manuscript describes a novel way to estimate mitochondrial respiratory chain complex V reverse activity, namely ATP hydrolysis by measuring acidification using the Seahorse flux analyzer. The manuscript is novel and interesting but for the benefit of the reader some amendments are warranted.

General comments: 1-Since this approach is novel it would be helpful to depict the pathway in a figure showing the principles and reactions, in addition to the table .

2-In the discussion it would also be of interest to compare with other methodologies such as measuring the release of Pi (which has been used to detect ATP hydrolysis in frozen tissue mitochondria).

Specific comments: 1-Elaborate why you preferred mitotracker deep red over other mitochondrial dyes.

2-Did you at some point include cytosolic Atpase inhibitors such as Qabain in your experiments? If yes comment if no explain why.

3- Would you be able to detect CV deficiency? would than ECAR be low?

4- Could different initial freezing methods (-20,-70, liq.N2) alter the results?

5- Was there any other specific treatment of the 96 well plates other than centrifugation to make the sample stick? Was there any protein left in the supernatant after the centrifugation?

Reviewer #2 (Comments to the Authors (Required)):

Mitochondria, the "powerhouse" of the cells, convert energy to ATP using complexes I to V (oxidative phosphorylation). The ATP synthase (Complex V; CV) uses the electrochemical proton gradient generated by the electron transport chain to synthesize ATP. Under certain conditions, CV can also reverse its rotation and hydrolyze ATP to re-adjust the potential difference between the mitochondrial matrix and the intermembrane space. To discover the implications of ATP hydrolysis in health and disease, the authors describe a standardized, direct, and efficient way to measure ATP hydrolysis activity in frozen samples (tissues and primary culture cells). By utilizing the Agilent Seahorse XF96 technology, they use the acidification rate as a proxy for ATP hydrolysis; then they further combine their ATP hydrolytic capacity measurement HyFS (which stands for Hydrolysis in Frozen Samples) with maximal respiratory activity in the same assay. In general the studies are well-performed and the method seems useful, but could be further confirmed by Native PAGE studies of ATP synthase to demonstrate the amount of fully assembled ATP synthase that is able to participate in hydrolysis, particularly after repeated freeze-thaw cycles.

Some comments and further remarks:

1) Frozen and thaw cycles can cause the CV subunits to disassemble with each other. This may not affect the enzymatic activity but will affect the overall acidification rate since the FO will be unable to pump H⁺ ions in response to ATP hydrolysis by the enzyme if the two parts are uncoupled. How to then explain the increase in acidification with increasing freeze-thaw cycles? Native PAGE to demonstrate the intactness of the ATP synthase would be helpful here to prove that freeze-thawing the samples does not diminish the amount of assembled ATP synthase or cause a breakdown of mitochondrial membranes that would prevent the separation of H⁺ ions across the mitochondrial inner membrane. If the native PAGE shows disassembly, it will be important for the authors to discuss the other possible sources of acidification. The assumption is that this assay cannot be used with isolated mitochondria since acidification of the extramitochondrial fluid would not be standardizable?

2) In Figures 1, "representative" examples are shown, but there are no group data reported; group data would strengthen the arguments, further supporting the validity of the assay.

3) Figure 1E is not clear. It is supposed to demonstrate a dose dependent sensitivity to oligomycin, but it seems as if the differential response appears after ATP/FCCP are added whereas after oligomycin is added, all the acidification values decrease to the same low amount.

4) Figure 2C, the amount of oligomycin added at 34 min should be based on the amount of the mitochondria in the assay. More mitochondria may need more oligo to shut off. It is not clear that this was tried.

5) Figure 3: To normalize the ATP hydrolytic activity by the amount of intact CV, a Native PAGE should be performed, not just an SDS PAGE immunoblot.

6) The ACMA assay using freshly purified Submitochondrial Vesicles (SMV) should be one good reference assay to test if this

new AR assay reflects the true H⁺ pumping difference between the samples (Caviston, T.L., Ketchum, C.J., Sorgen, P.L., Nakamoto, R.K., and Cain, B.D.(1998). Identification of an uncoupling mutation affecting the b subunit of F1F0 ATP synthase in *Escherichia coli*. FEBS Lett. 429, 201-206).

7) If the goal is to establish an easy method to test the differences in ability of tissues or cells to perform ATP hydrolysis in health and disease, the normal frozen tissue and the disease-related tissue should be run for the AR and OCR side by side, to determine if the new method is sensitive enough to distinguish them.

Reviewer #1 (Comments to the Authors (Required)):

This manuscript describes a novel way to estimate mitochondrial respiratory chain complex V reverse activity, namely ATP hydrolysis by measuring acidification using the Seahorse flux analyzer. The manuscript is novel and interesting but for the benefit of the reader some amendments are warranted.

We appreciate the reviewer for his/her positive feedback.

General comments:

1-Since this approach is novel it would be helpful to depict the pathway in a figure showing the principles and reactions, in addition to the table.

We have now added a scheme illustrating the different reactions. It is now included in the new Figure 1.

2-In the discussion it would also be of interest to compare with other methodologies such as measuring the release of Pi (which has been used to detect ATP hydrolysis in frozen tissue mitochondria).

We have now added a more elaborate discussion on other alternative methods used to measure ATP hydrolysis (second paragraph in the discussion).

Specific comments:

1-Elaborate why you preferred mitotracker deep red over other mitochondrial dyes.

The thank the reviewer for bringing out this very important point. When we first described respirometry in previously frozen samples (PMID: 32432379), we also proposed a method to express the respiratory rates by mitochondrial content. We tried different dyes such as mitotracker red (MTR), TMRE, and mitotracker deep red (MTDR). While TMRE and MTR were membrane potential dependent, MTDR was not. This different behavior allowed us to use MTDR in previously frozen samples as well as fixed samples where there is no membrane potential involved. We are including below some of the panels used in our previous publication to illustrate these results, referring to the original Figure panels used in PMID: 32432379

PMID: 32432379: Figure 6C. Fluorescence intensity in isolated mitochondria incubated in the presence of FCCP or calcium and stained with either MTR or MTDR shows the membrane potential dependency of MTR, but not MTDR.

PMID: **32432379**:

Supplementary Figure 5A-B.

A) Ins1 live cells stained with MTR or MTDR and imaged before (left panel), after fixation with 4% PFA (middle panel) and fixation plus

permeabilization with Triton X (right panel). (B) Hepatic cell line Hep2G stained with MTR or MTDR live and after fixation with PFA 4%. In both cases, MTDR stays after fixation in a similar pattern that before fixation

2-Did you at some point include cytosolic Atpase inhibitors such as Qabain in your experiments? If yes comment if no explain why.

We acknowledge the reviewer for this suggestion. We initially thought about including Ouabain as an inhibitor of other ATPases. However, we didn't perform those experiments since the cytosolic fraction didn't have any hydrolytic activity (Figure 3B)

3- Would you be able to detect CV deficiency? would than ECAR be low?

This a difficult question to answer since depending on the nature of the mutation we might be affecting not only activity but also stability of Complex V. In addition, if Complex V activity is affected, it could target synthesis, hydrolysis or both. We have tried ATP6 mutant (ATP6^{L156R}) and shown that ATP hydrolysis is increased in that mutation model (under review in EMBO J, EMBOJ-2022-111699). On the other hand, mutations in ATPIF1, a known modulator of Complex V activity can result in either increased or decreased hydrolysis (reviewed in PMID: **26876430**)

4- Could different initial freezing methods (-20,-70, liq.N2) alter the results?

This is a very interesting point. Previously, we have shown that leaving the samples at RT or 4 °C for 3h prior to freeze does not have any effect in maximal respiration (PMID: **32432379**), so we don't expect any changes in the hydrolysis capacity either.

However, since we are freezing the dry cell pellet in absence of any protease inhibitor, we do not recommend long-term storage at -20 °C. Initial freezing at -20 °C could be done, followed by long-term storage at -80 °C.

5- Was there any other specific treatment of the 96 well plates other than centrifugation to make the sample stick? Was there any protein left in the supernatant after the centrifugation?

We did not perform any treatment (PDL or Celltak coating) to the XF96 well plates. We followed the same protocol as for isolated mitochondria or respirometry in previously frozen samples (RIFS; PMID: **32432379**): centrifugation with no brake. When we established the protocol for RIFS, we tried different coating conditions to see if there were any differences in the samples sticking to the bottom of the plate or in the Seahorse signal. After trying different alternatives, we observed that coating the plates didn't make any differences in the assay.

Reviewer #2 (Comments to the Authors (Required)):

Mitochondria, the "powerhouse" of the cells, convert energy to ATP using complexes I to V (oxidative phosphorylation). The ATP synthase (Complex V; CV) uses the electrochemical proton gradient generated by the electron transport chain to synthesize ATP. Under certain conditions, CV can also reverse its rotation and hydrolyze ATP to re-adjust the potential difference between the mitochondrial matrix and the intermembrane space. To discover the implications of ATP hydrolysis in health and disease, the authors describe a standardized, direct, and efficient way to measure ATP hydrolysis activity in frozen samples (tissues and primary culture cells). By utilizing the Agilent Seahorse XF96 technology, they use the acidification rate as a proxy for ATP hydrolysis; then they further combine their ATP hydrolytic capacity measurement HyFS (which stands for Hydrolysis in Frozen Samples) with maximal respiratory activity in the same assay. In general, the studies are well-performed and the method seems useful, but could be further confirmed by Native PAGE studies of ATP synthase to demonstrate the amount of fully assembled ATP synthase that is able to participate in hydrolysis, particularly after repeated freeze-thaw cycles.

We thank the reviewer for his/her appreciation of our work.

Some comments and further remarks:

1) Frozen and thaw cycles can cause the CV subunits to disassemble with each other. This may not affect the enzymatic activity but will affect the overall acidification rate since the FO will be unable to pump H⁺

ions in response to ATP hydrolysis by the enzyme if the two parts are uncoupled. How to then explain the increase in acidification with increasing freeze-thaw cycles?

We didn't see any increase in the acidification rate with increase in freeze-thawing cycles but increase in the oligomycin sensitivity, due to the increase in the dynamic range of the assay.

Native PAGE to demonstrate the intactness of the ATP synthase would be helpful here to prove that freeze-thawing the samples does not diminish the amount of assembled ATP synthase or cause a breakdown of mitochondrial membranes that would prevent the separation of H⁺ ions across the mitochondrial inner membrane. If the native PAGE shows disassembly, it will be important for the authors to discuss the other possible sources of acidification.

This is a very interesting point. To validate that the amount of fully assembled ATP synthase is not compromised by the repeated freeze-thaw cycles, we compared two different native preparations (new Fig. 3K). One of them is a standard native preparation using primary fibroblasts (no FT), while the second one is a preparation performed after the cell pellet underwent four freeze-thaw cycles (4 FT cycles). The lack of changes between the two preparations demonstrates that, despite being advantageous for substrate accessibility, repeated freeze-thaw cycles do not impact CV assembly and, consequently, CV hydrolytic capacity. A new method section contemplating the native sample preparation and immunoblots has been added to the manuscript.

The assumption is that this assay cannot be used with isolated mitochondria since acidification of the extramitochondrial fluid would not be standardizable?

We have shown that this assay could be used in fresh and frozen isolated mitochondria (Figure 2). In these samples, the acidification detected comes from Complex V working in reverse after inhibition of the ETC with antimycin and injection of ATP to trigger the enzymatic reaction.

2) In Figures 1, "representative" examples are shown, but there are no group data reported; group data would strengthen the arguments, further supporting the validity of the assay.

We acknowledge the reviewer for his/her observation. We decided to show representative traces in this figure (now new Figure 2) because we found them more explicative and helpful to follow the new method and its limitations, especially in mitochondria samples which are not the main target sample for the method described. In the following figures, showing method optimization and validation, group results are always shown. Additionally, it is difficult to provide group data with values coming from different mitochondrial preparations without having an internal control run in parallel in all the experiments.

Despite the same method is always used for mitochondria isolation, the absolute values obtained may vary from one day to another.

That said, the reviewer's point is very valid, so to strengthen the results shown we are providing here a second (sometimes third) replica of the acidification rates from panels Fig. 2B, 2C, 2D, and 2G (see Fig.1 for reviewers below). We would be happy to include this figure as supplementary information if the reviewer considers this would be a helpful addition.

Fig. 1. Additional replicas from some of the experiments shown in main Figure 2. (A) Acidification rates from fresh mitochondria extracted from heart. (B) Acidification rates from frozen mitochondria extracted from heart. (C) Acidification rates from fresh mitochondria extracted from primary fibroblasts. (D) Acidification rates from frozen mitochondria extracted from primary fibroblasts.

3) Figure 1E is not clear. It is supposed to demonstrate a dose dependent sensitivity to oligomycin, but it seems as if the differential response appears after ATP/FCCP are added whereas after oligomycin is added, all the acidification values decrease to the same low amount.

We acknowledge the reviewer for this comment. We have now clarified this point in the text and explained better the graph. The oligomycin conditions described in the legend of the graph correspond to the composition of the MAS added to the samples after centrifugation (since the beginning of the assay) and not to the port injection. The oligomycin injected in port D is used as a control of fully inhibition of ATP hydrolysis.

4) Figure 2C, the amount of oligomycin added at 34 min should be based on the amount of the mitochondria in the assay. More mitochondria may need more oligo to shut off. It is not clear that this was tried.

When performing the assay, we normally add higher concentrations of oligomycin to account of total inhibition of CV hydrolysis. We have shown in former Figure 1 (new Figure 2) that oligomycin at that concentration works in isolated mitochondria. Using the same amount of oligomycin in lysates, where mitochondria content is diluted, we expect the maximal inhibition with the concentration used.

5) Figure 3: To normalize the ATP hydrolytic activity by the amount of intact CV, a Native PAGE should be performed, not just an SDS PAGE immunoblot.

This is a very valid point. We have now performed BNAGE analysis of Complex V assembly in homogenates from liver and heart and included in the new Figure 4H. A new section in Table 2 has been added to account for this additional normalization method.

6) The ACMA assay using freshly purified Submitochondrial Vesicles (SMV) should be one good reference assay to test if this new AR assay reflects the true H⁺ pumping difference between the samples (Caviston, T.L., Ketchum, C.J., Sorgen, P.L., Nakamoto, R.K., and Cain, B.D.(1998). Identification of an uncoupling mutation affecting the b subunit of F1F0 ATP synthase in Escherichia coli. FEBS Lett. 429, 201-206).

We considered using ACMA to further validate our assays but that would involve isolating vesicles from cells (the assay is described in inverted membrane vesicles) and it would also involve coupling it to other chemical reactions such as the phosphorus reaction/molybdate reaction (PMID: **2874137**; PMID: **13061491**), rather than measuring the ATP hydrolytic activity directly as we propose.

To further verify the validity of our approach, we have add a third validation approach where we have partially silenced ATP1F1 using shRNA and measured HyFS. These results have been now included in the revised Figure 5. The data show that a 50% reduction in ATP1F1 expression is enough to double the amount of ATP hydrolytic capacity in comparison to a scramble control (Fig. 5G-J).

7) If the goal is to establish an easy method to test the differences in ability of tissues or cells to perform ATP hydrolysis in health and disease, the normal frozen tissue and the disease-related tissue should be run for the AR and OCR side by side, to determine if the new method is sensitive enough to distinguish them.

We acknowledge the reviewer for this comment. The ultimate goal of this assay is the validation of the technique in health and disease, as we did previously in for our work in respirometry in previously frozen samples (RIFS) (PMID: **32432379**). We have validated HyFS in health and disease in models of mitochondrial dysfunction such as primary human fibroblasts with electron transport chain deficiencies as well as muscular dystrophy models where mitochondrial function is impaired (under review in EMBO J, EMBOJ-2022-111699).

November 22, 2022

RE: Life Science Alliance Manuscript #LSA-2022-01628R

Prof. Orian S. Shirihai
University of California, Los Angeles
Department of Molecular and Medical Pharmacology, David Geffen School of Medicine
650 Charles East Young Drive South
CHS 27-200
Los Angeles, CA 90095

Dear Dr. Shirihai,

Thank you for submitting your revised manuscript entitled "A Novel Approach to Measure Complex V ATP Hydrolysis in Frozen Cell Lysates and Tissue Homogenates". We would be happy to publish your paper in Life Science Alliance pending final revisions necessary to meet our formatting guidelines.

- please include the new panels provided as additional examples for fig. 2 in a supplementary figure as suggested by Reviewer #2
- please consult our manuscript preparation guidelines <https://www.life-science-alliance.org/manuscript-prep> and make sure your manuscript sections are in the correct order
- please add ORCID ID for secondary corresponding author-they should have received instructions on how to do so
- please add the Twitter handle of your host institute/organization as well as your own or/and one of the authors in our system
- please use the [10 author names, et al.] format in your references (i.e. limit the author names to the first 10)
- please add a callout for Figure 4F to the main manuscript text
- please add a legend for your tables to the figure legend section and make sure your table files are uploaded as an editable doc or excel file or are in the doc file of your main manuscript

Figure Check:

- Figure 4J: scale bars should be more visible

A. FINAL FILES:

-- Summary blurb (enter in submission system): A short text summarizing in a single sentence the study (max. 200 characters including spaces). This text is used in conjunction with the titles of papers, hence should be informative and complementary to the title. It should describe the context and significance of the findings for a general readership; it should be written in the

present tense and refer to the work in the third person. Author names should not be mentioned.

B. MANUSCRIPT ORGANIZATION AND FORMATTING:

Sincerely,

Reviewer #1 (Comments to the Authors (Required)):

My comments comments have been adequately addressed and the revised MS is much improved.

Reviewer #2 (Comments to the Authors (Required)):

The manuscript provides interesting use of known techniques to measure acidification rate as a read-out of ATP synthase in hydrolysis mode. The authors have done a wonderful job of discussing the caveats and normalization techniques and showing how some of these are inadequate. The authors have substantially improved the manuscript. I think the panels provided as additional examples for fig. 2 can be placed in a supplementary file but if the journal does not allow supplementary data, these new panels can be omitted. This reviewer is satisfied with what has been shown in the main figures.

December 1, 2022

RE: Life Science Alliance Manuscript #LSA-2022-01628RR

Prof. Orian S. Shirihai
University of California, Los Angeles
Department of Molecular and Medical Pharmacology, David Geffen School of Medicine
650 Charles East Young Drive South
CHS 27-200
Los Angeles, CA 90095

Dear Dr. Shirihai,

Thank you for submitting your Methods entitled "A Novel Approach to Measure Complex V ATP Hydrolysis in Frozen Cell Lysates and Tissue Homogenates". It is a pleasure to let you know that your manuscript is now accepted for publication in Life Science Alliance. Congratulations on this interesting work.

DISTRIBUTION OF MATERIALS:

Again, congratulations on a very nice paper. I hope you found the review process to be constructive and are pleased with how the manuscript was handled editorially. We look forward to future exciting submissions from your lab.

Sincerely,
